# Localized TEC enhancements in the Southern Hemisphere

Ilya K. Edemskiy[1]

[1]Institute of Atmospheric Physics, Czech Academy of Sciences, Prague, Czech Republic

*Correspondence to*: Ilya K. Edemskiy (edi@ufa.cas.cz)

**Abstract.** The paper is dedicated to investigation of localized TEC (total electron content) enhancements (LTEs), detected in the Southern Hemisphere via analysis of global ionospheric maps. Using data for different years (2014, 2015, 2018) we show presence of LTE almost independently on solar activity. It is shown as well that LTE is a phenomenon which can be observed serially: at the same universal time (UT) similar enhancement can manifest itself during several days. Intensity of LTEs varies in dependence on solar flux and does not directly depend on interplanetary magnetic field orientation; they occur under both geomagnetically disturbed and quiet conditions. The highest LTE occurrence rate is observed during period of local winter (April-September) for all analyzed years. The longest observed LTE series was detected during 2014 and lasted 80 days or 120 days if we exclude 2 daily gaps.

## 1 Introduction

The Southern Hemisphere (SH) ionosphere has not been investigated so broadly as one of the Northern Hemisphere (NH). Historically, most of the geophysical observations and measurements have been made to the north of the equator. Even now, having lots of observatories all around the globe, we have a lack of ground-based observations for a larger part of the Southern Hemisphere since it is mostly occupied by ocean. Satellite measurements allow us to investigate ionosphere over oceans but due to its high variability and the movement of satellites it is very difficult to observe the same region in the same conditions.

It is known that Southern Hemisphere contains some anomalous regions. South Atlantic Magnetic Anomaly (SAMA) is formed by a configuration of geomagnetic field which has a global minimum of intensity over South Atlantic and South America and makes it easier for energetic particles of inner radiation belt to precipitate, thus increasing ionospheric conductivity over the region (Abdu et al., 2005). South of the SAMA, in the south-eastern Pacific and South Atlantic Antarctic regions, combination of the geomagnetic field features and thermospheric winds produces an inverted diurnal plasma density pattern at equinoxes and in SH summer (October-March): the nighttime maximum is larger than the daytime minimum, and the phenomenon is known as the Weddell Sea Anomaly (WSA) (Horvath, 2006). Jakowski et al. (2015) showed that during periods of low solar activity in Asian longitudinal sector of SH it is possible to observe so called nighttime winter anomaly (NWA), when values of electron concentration are higher in winter than in summer. At the same time Yasyukevich et al. (2018) showed that winter anomaly manifests itself much less intensively in SH that in NH. It is

quite clear that due to these anomalies the structure and dynamics of ionosphere in both hemispheres should be different and should be investigated separately.

The most widely used and generally accepted the International Reference Ionosphere (IRI) empirical model (e.g., Bilitza, 2018) does not predict some features of the SH ionosphere sufficiently. Karia et al. (2019) analyzing predictions of IRI-2016 showed that the model does reproduce the observed NWA effect, though at a different longitude and could be improved for

better predictions. Comparing TEC measurements and results of IRI-PLAS, Alcay and Oztan (2019) found that in SH the model generally overestimates the GPS-TEC measured at stand-alone stations with the maximal difference about 15 TECU. Karpachev and Klimenko (2018) proposed a new model reproducing the structure of the high-latitude ionosphere more accurately than IRI-2016 and noted that inaccuracies of IRI in that region are connected with inaccuracy of ground-based sounding data, which varies during a day. However, none of these models predict the occurrence of localized enhancements

of electron concentration especially in Southern Hemisphere.

The most typical irregularities in distribution of electron concentration are produced during geomagnetic storms. Foster and Coster (2007) investigating storm enhanced densities (SED). They showed that during severe and extreme storms it is possible to detect SEDs which in maps of total electron content (TEC) could be observed as localized TEC enhancements (LTE). The authors showed that during a storm recovery phase LTEs can be detected in the night side ionosphere at the

middle latitudes of both hemispheres, in magneto-conjugated regions. The authors note that the observed enhancements are approximately corotating in place over the positions in which they were formed earlier in the event. However, the LTE phenomenon studied by Foster and Coster (2007) is different from the LTE phenomenon studied by us. During analysis of ionospheric response to geomagnetic storm of 15 August 2015 Edemskiy et al. (2018) detected a curious LTE in global ionospheric maps (GIMs). Unlike the LTEs observed by Foster and Coster (2007) this enhancement was observed in sunlit

(near-noon) area of the Southern Hemisphere and lasted for several hours. It was not corotating but changing position following the Sun and propagating along the geomagnetic parallels. Using quite a simple detection algorithm Edemskiy et al. (2018) found about 30 similar events in the Southern Hemisphere during 2010-2016 and the most of the detected LTEs were observed during relatively disturbed periods. The authors showed direct dependence of number of the detected LTEs on solar activity level and suggested that the generation of the enhancements is connected with the orientation of interplanetary

magnetic field (IMF), namely with Bz.

The present article is an attempt to detect more LTEs developing in Southern Hemisphere during different solar activity periods and to investigate them more carefully trying to understand mechanisms of their generation. Section 2 describes data and methods, section 3 presents results, section 4 deals with discussion and possible mechanism, and section 5 summarizes main results.

## 60 2 Data and methods

The algorithm used by Edemskiy et al. (2018) had some disadvantages. The used fixed detection threshold did not allowed them to detect relatively weak LTEs. The applied comparison with a weekly TEC median excluded from the consideration

possible series of such formations. Trying to improve the effectiveness of LTE detection we introduced following criteria for the TEC formation. In this paper a TEC enhancement is considered as LTE if it is:

- located in middle latitudes of sunlit region. Mainly we investigate LTEs, which are clearly observed in Indian and Southern part of Atlantic Oceans and do not take into account enhancements in Northern Hemisphere. At the same time, LTEs in SH are not accompanied by any LTE in NH and such a focusing on SH LTEs is quite reasonable.

- spatially limited by relatively lower TEC values. Normalized difference between squared maximal value in LTE and minimal one at its border ($\Delta = 1 - (I_{edge}/I_{max})^2$) should be no less than 20%. Generally that means that between southern

crest of equatorial ionization anomaly (EIA) and a region of enhanced TEC there should be observed a clear trough.

- confined and have a border of lower TEC values ($\Delta \geq 20\%$) no farther than in 40° in longitude from the location of maximal TEC value. Mainly that means that we do not consider longitudinally stretched enhancements assuming different mechanism of their generation.

These criteria were applied to analysis of global ionospheric maps (GIMs). Currently, these maps are provided by several

scientific groups: CODE (codg), ESA (esag), JPL (jplg), UPC (upcg), Whuan university (whug), Chinese Academy of Sciences (CAS - casg). IGS service also provides maps (igsg) created as a combination of maps from CODE, UPS, ESA and JPL. The spatial resolution is 2.5°x5° in latitude and longitude, respectively, and temporal one is 2 h (1 h for CODE maps since 2015). Maps are calculated from slant (sTEC) values measured at 200-350 GNSS receivers (in dependance on the data availability and the used method) all around the world with application of some interpolation method. Global ionospheric

maps from all the above mentioned groups are freely available at CDDIS server (ftp://cddis.gsfc.nasa.gov/gps/products/ionex). According to Roma-Dollase et al. (2018) CODE and CAS maps have the lowest relative errors in the South Atlantic and Indian Ocean regions. Taking into account the high temporal resolution of CODE maps and more clear information about the used data in the headers of these maps, we use CODE GIMs in the present paper.

To confirm an LTE presence we use measurements from SWARM and COSMIC satellite missions. The SWARM mission was launched by ESA at the end of 2013. It is mainly aimed to investigation of Earth's magnetic field. The mission includes three satellites at polar orbits of about 500 km (460 km for Alpha and Charlie, and 530 km for Bravo). The data are available via browser-based application (https://vires.services/) or via API tool (https://github.com/ESA-VirES/VirES-Python-Client). In the present paper SWARM in-situ measurements of electron density are used.

The project COSMIC (Constellation Observing System for Meteorology Ionosphere & Climate) provide measurements of upper atmosphere and ionosphere parameters. In the present paper we use TEC profiles obtained via radio occultation (RO) receiving of GPS signals. To distinguish this data from the standard ground-based TEC measurements, we use abbreviation SS TEC (satellite-to-satellite TEC). COSMIC data is freely provided as NetCDF files (https://cdaac-www.cosmic.ucar.edu/). We analyze mainly the occurrence rate of LTE and its dependence on space weather. Quantitative analysis of LTEs

generally consists of definition of maximal TEC value over the investigation region and calculation of its relation to mean

TEC value over the region. Analysis of the dependence of these parameters on near space conditions was made during the investigation. LTE shapes vary widely and are quite difficult for formalization.

To analyse connection of the observed features of ionospheric dynamics with geomagnetic field we use SuperDARN altitude adjusted corrected geomagnetic coordinates (AACGM) (Shepherd, 2014) as a Python module developed by Angeline Burrell

(https://github.com/aburrell/aacgmv2). To create maps in geomagnetic coordinates we place each TEC cell from GIM map at the corresponding magnetic latitude and longitude calculated with AACGM for an altitude of 100 km.

Files of GIMs in IONEX format were treated with the python package GNSS-LAB created by Ilya Zhivetiev (https://github.com/gnss-lab). Processing and presentation of data were made with Python libraries Numpy (https://numpy.org) and Pandas (https://pandas.pydata.org/). Geomagnetic indices (Kp, Dst, AE, etc.) and other parameters

of near space (including $F_{10.7}$ index for estimation of solar activity) were taken from OMNI database (https://omniweb.gsfc.nasa.gov). During the investigated years monthly averaged $F_{10.7}$ values were varying in ranges 130-160 (2014), 95-135 (2015), 65-75 (2018) sfu corresponding to high, relatively high and low level of solar activity. It should be noted that AE index values during 2018 are available only for January and February both in OMNI and Kyoto WDC (http://wdc.kugi.kyoto-u.ac.jp/dstae/index.html) databases.

**3 Results**

An example of a clearly observed LTE was detected at April 5, 2014 (fig. 1). The disturbance reached the highest intensity in a period 10-12 UT when TEC values in a most intense part of the disturbance exceeded 78 TECU. This value is comparable to equatorial TEC values. The highest values were detected in a latitudinal region 45-70°S. At the same time, TEC values of the entire region (30-70°S, 0-90°E) were enhanced.

It is possible to distinguish two parts in presented LTE: midlatitudinal (MLTE) and subpolar (SLTE). The LTE of April 5 has strong subpolar part and weaker but still pronounced midlatitudinal one. As it will be shown later, such a strong SLTE is not typical and in some cases it is not detected at all. However, both MLTE and SLTE were presented during this even quite clearly for several hours and that was the main reason to describe this particular case in more details.

During its development the LTE changes latitudinal position in a range 30-80°S corresponding to range of geomagnetic

parallels within 35-70°S (red lines in fig. 1A). Phases of the development during April 5 are shown in geomagnetic coordinates (AACGM) in panels B-K in fig. 1. As it could be seen from the figure, the LTE exists during the entire day and changes its intensity unevenly. The less intensive MLTE part persists for a longer time and has lower magnitude than the brighter SLTE. Both parts are confined in their own ranges of geomagnetic latitudes: 30-50°S (MLTE) and 50-65°S (SLTE). During the whole period shown their positions keep approximately the subsolar area (local noon).

It is necessary to say that the LTEs are detected most clearly over Atlantic and Indian oceans, where amount of GNSS stations is insufficient. White squares in fig. 1 mark location of the receivers providing CODE with data for TEC maps. Only a few are located in ocean (on islands) and the only is in 30-60°S latitudes of Indian Ocean (Kerguelen Islands, KERG). Therefore LTE detection has to be confirmed by other observations.

In-situ measurements of electron concentration Ne from SWARM satellites allow us to validate TEC distribution presented
by GIM. Left panel of figure 2 presents Ne values, observed during 8-14 UT at April 5, 2014. Each track is marked by a
colored dot corresponding to satellite: Alpha (red), Bravo (blue) and Charlie (cyan); digits of the corresponding color marks
the satellite position at the the beginning and the end of the track in a format HHMM (hours and minutes). All the satellites
were moving from equator to pole.

The area of extremely high concentration of electrons is clearly observed in data from all the three satellites. Blank areas in
measurements from Alpha and Charlie during 11:30-13:30 mark the zone of concentrations exceeding color axis limitation.
Temporal differences between passages of the satellites allow us to observe the dynamics of the LTE. The most intensive
part is shown by Alpha's measurements. Charlie is ahead of Alpha by about 15 minutes and 2.5° of longitude and its
measurements in general show lower concentration especially for a period 8-12 UT. Most probably such a difference is
caused by movement of the enhancement: according to the GIM LTE is located in subsolar region and follows the Sun.
Bravo is about 30 min and 12° behind Alpha and its measurements shows significantly lower concentration than the other
satellites. It could point not only to the disturbance displacement but also to its distribution with altitude, since the orbit of
Bravo is 70 km higher than those of Alpha and Charlie.

The distribution of electron concentration with altitude can be analyzed using radio occultation measurements by COSMIC
satellites. Profiles of SS TEC during April 5 are presented in the right panel of fig. 2. Each SS TEC value in a profile is
obtained on a bent satellite-to-satellite ray and is attributed to a tangent point of the ray (Rocken et al., 2000). Projections of
the tangent points during each profile measurement are shown in left panel of fig. 2 with the same color as the profile. Cross
marks on the trajectories and nearby digits indicate location and time of the lowest altitude measurement (last measured
value before GPS satellite occultation). Due to the phenomenon of LTE series which will be described later, TEC values
over the given region are enhanced almost during the whole month. To demonstrate ionosphere profile without any
enhancement in GIM we have chosen October 19, 2014. The profile measured by COSMIC at 10:12UT is shown by dashed
blue line in fig. 2.

It is quite clear that the detected disturbance was propagating according to solar motion and had the highest electron
concentration in F region at about 11 UT. Profiles also show that electron concentration at an altitude 460 km could be 1.5-2
times higher that at 530 km, which is in a correspondence with SWARM measurements.

LTEs similar to the one detected on April 5 could be observed during several days in a row. In the particular case of April
2014, LTEs southward of Africa were detected since March 18 till April 11. TEC maps at 10:00 UT for April 1-9, 2014 are
presented at the left side of fig. 3. The geomagnetic conditions during this period were slightly disturbed: maximal value of
Kp was 4 (April 7), and minimal Dst value was about -25 nT (April 7-8). The intensity and shape of the presented LTEs vary
from day to day, but at the same UT all of the LTEs occupy the same region. Intensities of MLTE and SLTE vary
independently. SLTE is more intense only on April 5. Mostly its intensity is either close to that of MLTE (April 1, 3, 4, 7, 9)
or lower (April 2, 6 and 8). We define such a continuous sequence of LTEs observed day by day as a series of LTE. At least
two consistently observed LTEs are considered as a series. The other panels of fig. 3 demonstrate LTE series observed

during years of relatively high (2015, in a middle) and low (2018, at the right) solar activity. The activity level was was estimated $F_{10.7}$ index values. Intensity of the observed LTEs varies according to global electron content, which depends on solar activity (e.g., Afraimovich et al., 2008). Disturbances of 2015 still have two different zones of LTEs, while all the presented LTEs of 2018 apparently are of MLTE type (see April 6 in fig. 3, left). Geomagnetic activity during the presented days was from moderate to low and there was no clear correlation between indices (Kp, Dst, etc.) and shapes or intensities of the disturbances.

The series of LTE were detected during all the three years. Figure 4 shows variations of solar ($F_{10.7}$) and geomagnetic (AE and Dst) indices during each year indicating days with LTE detected (blue and red bars). All the indices are taken daily averaged. According to the figure the most often LTEs are detected in autumn and at the beginning of winter (since March till June-July). Speaking of the series, the absolute maximum of their occurrence is observed in autumn-winter period as well with the longest ones during April-June. In late spring and in summer no LTE series were usually observed. The most interesting series here lasted 80 days of 2014 from May to July (fig. 4, top). It is possible to see that only several short gaps separate this series from two others in autumn and probably the entire period of late March-July should be considered to include one long series. Such a long sequence occupying the third part of a year definitely points to some regular process. For the other years the same season contains majority of the LTE series, but separated with more frequent and wider gaps. It is interesting to see that during a year of low solar activity (2018) we detect more series than during a moderately active one (2015).

Red bars in fig. 4 mark the days when the intensity of SLTE was higher than the intensity of the accompanying MLTE (as in fig. 1). Such bright SLTEs were detected only during years of relatively high solar activity (2014, 2015). Comparing their occurrence with the averaged indices we can hardly observe clear dependence between the detection of them and conditions of the near space.

Due to large variety of spatial forms and intensity distributions of LTEs (fig. 2) it is not easy to select a key parameter for an analysis over three years. We simplified the task by analyzing variations of maximal TEC (TECmax) value observed at 10 UT in a region 30°W-60°E, 30-60°S. Panels of figure 5 present distributions of TECmax during the entire three years versus main parameters of the near space: solar flux at 10.7 nm ($F_{10.7}$, (a, e)), By(b) and Bz(f) components of IMF, geomagnetic indices SYM-H(c) and Dst(g). Relative intensity of an LTE could be analyzed by a ratio of TECmax to average TEC over the region (TECratio). Distributions of TECratio versus the main parameters (not presented) are quite chaotic and do not demonstrate any pronounced dependence, except of AE (fig 5d) and IMF intensity B (fig 5h). The last ones do not show a clear dependence as well, but it is possible to see that TECratio values tend to be higher with increased AE and B values. It was found that during active years (2014-2015) maximal value quite clearly depends on $F_{10.7}$ index (fig 5a). It was not a surprise since maximal value directly depends on the entire amount of electrons in ionosphere, which is driven by solar radiation. Speaking of all the other parameters, we can hardly see any specific dependence on them.

## 4 Discussion

Being observed separately SH LTEs were previously supposed to be a relatively rare phenomenon produced by some specific condition of near space (Edemskiy et al., 2018). However the data presented above showed that LTEs occur quite often and can be observed in a sequence during a relatively long period when geomagnetic conditions and solar parameters vary significantly. The presented distributions did not reveal any pronounced dependence except the one between maximal TEC value in the region and solar flux intensity (fig. 5a). Obviously TECmax linearly depends on total amount of electrons in ionosphere or global electron content and the last one is known to be dependent on $F_{10.7}$ index (e.g., Astafyeva et al., 2008). At the same time it is surprising that the other distributions in fig. 5 do not show clear dependence on near space parameters. The previous suggestion (Edemskiy et al., 2018) of SH LTE occurrence only during disturbed conditions and especially with the observed negative Bz appears not to be entirely correct.

Being detected at 10 UT and occupying the same region of Southern Hemisphere all the observed LTEs show wide variety of shapes making it difficult to classify them. At the same time MLTE and SLTE intensities apparently independent and that can aim to different mechanisms of formation and that could be used as a classification. We selected cases of bright SLTE (fig. 4, red bars) and calculated separately distributions of TECmax and TECratio versus AE, Bz and SYM-H for these days (fig 6, top) and for all the other LTEs (fig 6, bottom) detected in SH over the investigated years.

The figure shows that most of the bright SLTEs were detected at the moments of negative Dst and SYM-H, and high values of AE index. In total that means that bright SLTEs are often observed during disturbed geomagnetic conditions. It is known that SEDs generated in high latitudes during geomagnetic storms could be observed in TEC maps as localized enhancements (e.g. Foster and and Rideout, 2007) and the detected SLTE could be a manifestation of some SED.

According to Foster (2008) SEDs are typically observed during severe geomagnetic storms and generally are formed by a F-region plasma driven upward and poleward (ExB direction) by eastward electric field penetrated into the inner magnetosphere at the early phase of a geomagnetic storm. Being formed by the fountain effect the enhanced plasma of EIA peaks can be redistributed during extreme events when uplifting plasma reaches higher-latitude flux tubes, resulting in enhanced electron density near the plasmapause. Most often such uplifts are observed in the dusk sector (Foster, 2008). Further development of the event can lead to generation of sub-auroral polarization stream creating SED as a connection between dusk sector and a region of dayside cusp. So partially the detected SLTEs could be generated via the described mechanism.

At the same time several features of SLTE should be highlighted. First, intense SLTEs were detected during a relatively quiet period as well. At least a quarter of them were detected with AE index values lower that 200 nT (fig. 6a). Second, mostly SEDs are believed to be plume-shaped, clearly connected to EIA region, and have high intensities along the entire plume. The used criteria excluded from consideration both the stretched formations and the ones having connection to EIA. So not all of the SLTEs are produced by some kind of SEDs and even if they are, the mechanism of their generation should differs from the one in NH.

Measurements of electron concentration by SWARM clearly confirm presence of SLTE, when in middle latitudes only generally enhanced Ne values are presented (fig. 2) without clear maximum of MLTE. This apparent absence of MLTE should be explained by orbit position: all the flights of the satellites at Apr 5 were crossing MLTE at the east edge (at 10UT it was about 70°E ) where TEC falls and do not show significant peak. Due to the orbital motion SWARM satellites appears over the same region at different time and at some moments it is possible to see an exact intersection of a LTE. During Apr 18, 2014 a pronounced LTE was observed both in GIM and in Ne measurements (figure 7). That is quite clear from the figure that enhanced concentration is observed at both altitudes of A/C (460 km) and B (530 km) satellites orbits. Together with the data shown in fig. 2 it makes a good point to believe that LTEs of both types (MLTE and SLTE) are predominantly located in the F2 region.

Midlatitudinal LTEs are mostly detected in the same region of SH (at 10 UT), but demonstrate wide variety of shapes. It is difficult to say that their generation is driven by space weather since no clear dependence on its main parameters were found for both the occurrence rate and the intensities of LTEs. Most probable the mechanism of their formation is connected to some kind of plasma redistribution since the most often the enhancements are observed during autumn-winter period (April-August, fig 4) when intensity of solar ionization in middle latitudes should be less effective than during summer. Apparently the mechanism is not connected with or not organized like the fountain effect since  last one typically gives a quasi-symmetrical (with respect to equator) pattern, and similar LTEs were not detected in magneto-conjugated region of NH. Moreover the intensities of TEC in corresponding part on NH during LTE detection are typically lower than ones in SH. Together with seasonal asymmetry that reminds winter anomaly (WA) phenomenon: F2-layer density values are greater in the winter hemisphere than in the summer hemisphere. It should be noted that, using COSMIC RO data Gowtam and Tulasi Ram (2017) showed that at altitudes within 300-700 km WA effect is confined only to morning-noon hours and only to low-latitudes, claiming absence of WA in middle latitudes. Yasyukevich et. al. (2018) analysing GIM and satellites' data confirmed that SH WA is much less pronounced than NH WA and the region of its observation is mostly located in the southern part of Indian Ocean. The authors also showed dependence of the anomaly intensity on solar activity and claimed that it could be observed only during high solar activity years. Moreover, they concluded that in TEC the anomaly could be observed only in periods with  $F_{10.7} > 170$ SFU. As it was shown above only intensity of LTE depends on $F_{10.7}$ , but not the occurrence rate and only few of them were detected during periods of such a high values of $F_{10.7}$ index. Higher TEC values in SH are observed during really low   $F_{10.7}$ (entire 2018) as well. So the mechanism of LTE generation probably is not connected with winter anomaly.

Being observed dynamically LTEs show development along geomagnetic parallels within 30-70°S of geomagnetic latitude, approximately in boundaries of magnetic shells L = 2-4 (fig. 1 B-K), and could be observed permanently for several days with slight changes of their form and intensity.  Anderson et al. (2014) detected hotspot of energetic electron precipitation E > 300 keV at SH at geomagnetic latitudes 55–72°S (much less pronounced at NH) and geographic longitudes 150°W–60°E. However, this result is based on nighttime observations, i.e. predominantly autumn-winter observations, when almost no LTEs have been detected. Using POES data for analysis of South Atlantic Anomaly, Domingos et al. (2017) found a plume

of particle flux located within L=2.5-3 in South Atlantic. The position of the plume was in a good correlation with a typical LTE position. However, the plume was observed in December when occurrence rate is minimal (fig. 4). Moreover, for the LTE analyzed in detail by Edemskiy et al. (2018) it was shown that particle precipitations are not responsible for that LTE. So most probably LTEs are not directly connected with the increased fluxes.

Statistically electron concentration over the western part of Indian Ocean is enhanced during equinox periods. Jee et al. (2009) investigating TOPEX data over 1992-2005 showed that during March-April noontime TEC values are significantly increased over the southern part of Africa and its Indian Ocean shore. Similar increment with lower intensity is shown during September-October. At summertime it is still possible to observe this enhancement with much less intensity. In winter the region of enhanced TEC depends on solar activity: during high activity period no enhancement is observed. During low activity the Equatorial Anomaly area grows, reaching 30ºS over Africa and TEC values at south of Africa are increased as well. Our results show higher probability of wintertime LTE detection during lower activity years. Analyzing GIMs for 1998-2015 Lean et al. (2016) found typically enhanced TEC over the region during 10-16UT and according to data for 2000-2002 the highest values during March-May.

As a conclusion we should say that at the present moment the generation mechanism is still unclear for us. The phenomenon of SH LTE is observed quite regularly, in periods of different solar activity and under different conditions of near space manifesting itself even during geomagnetically quiet periods. Since we did not detect symmetrical phenomena in Northern Hemisphere we could conclude that the enhancements are a feature of Southern Hemisphere ionosphere and therefore they should be driven by combination of its specific conditions: geomagnetic field, oceanic ionosphere and system of winds. Such a regular phenomenon should be taken into account by models as well. Currently it is difficult to say if it is reproduced by models, since it is not well described in the literature. We could mention a paper by Lee et al. (2011) who showed presence of enhanced electron concentration formation over western part of Indian Ocean using measurements from GRACE and CHAMP satellites. The authors concluded that 2001 and 2007 IRI models did not predict the observed enhancement at all. So the phenomenon should be investigated more precisely since it will surely give us more clear understanding of global distribution of ionospheric plasma.

**5 Conclusions**

The paper shows that localized TEC enhancements in Southern Hemisphere are observed quite regularly and can be detected serially. Having clear seasonal asymmetry of occurrence they do not show any pronounced dependence on space weather parameters. Enhancements can be detected during both disturbed and relatively quiet geomagnetic periods with different level of solar activity. Midlatitudinal and subpolar LTEs seem to have different mechanism of generation and should be investigated separately in more details. At least half of the observed SLTEs were detected during disturbed conditions and could be connected with SED structures. At the same time, part of them occurred during relatively quiet conditions and that means that even generation of SLTEs can be driven by several different mechanisms. Midlatitudinal LTEs are observed more regularly and show a big variety of shapes and intensities. TEC values during MLTE detection are typically higher than

ones in conjugated region of HN. Absence of clear dependence of MLTEs occurrence rate on space weather makes it difficult to propose any certain mechanism of their generation.

The presented data lead us to the opinion that despite the observed LTEs were supposed to be an ionospheric disturbance they most probable are a feature of the Southern Hemisphere ionosphere. The phenomenon should be investigated in more details with some additional methods including comparison with different models of ionosphere.

**Acknowledgements**

The author is grateful to Jan Laštovička for the idea of the investigation, for fruitful discussions and corrections of the text. Author thanks Martin Pačes for his kind help with obtaining SWARM data and Nikolay Zolotarev for his help with COSMIC data treatment. Thanks to all the data centers, which provided data: NASA's Crustal Dynamics Data Information System (CDDIS), CODE scientific group, SWARM and COSMIC mission staff. Author grateful to the reviewers which
impact helped to improve the paper.

**Code/data availability**

All the used data are available in accordance to links in Data and methods section.

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

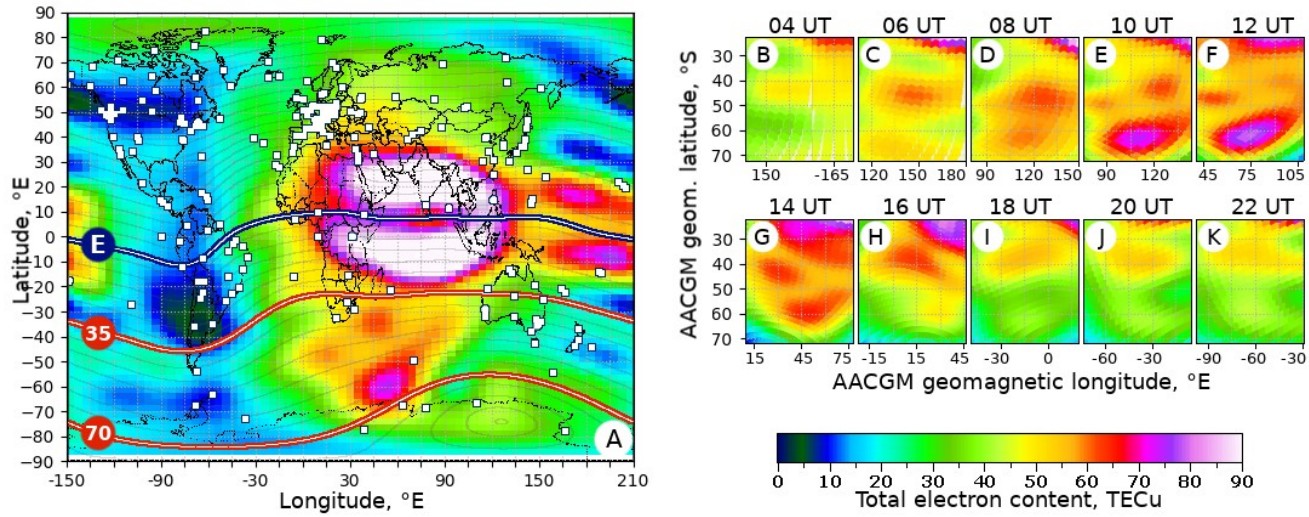


**Figure 1: Intense LTE observed at 10:00 UT of 05.04.2014 near Antarctica (A) and its development during a day in geomagnetic coordinates (B-K). LTE develops along geomagnetic parallels in a region 35-70°S (A, red lines)**

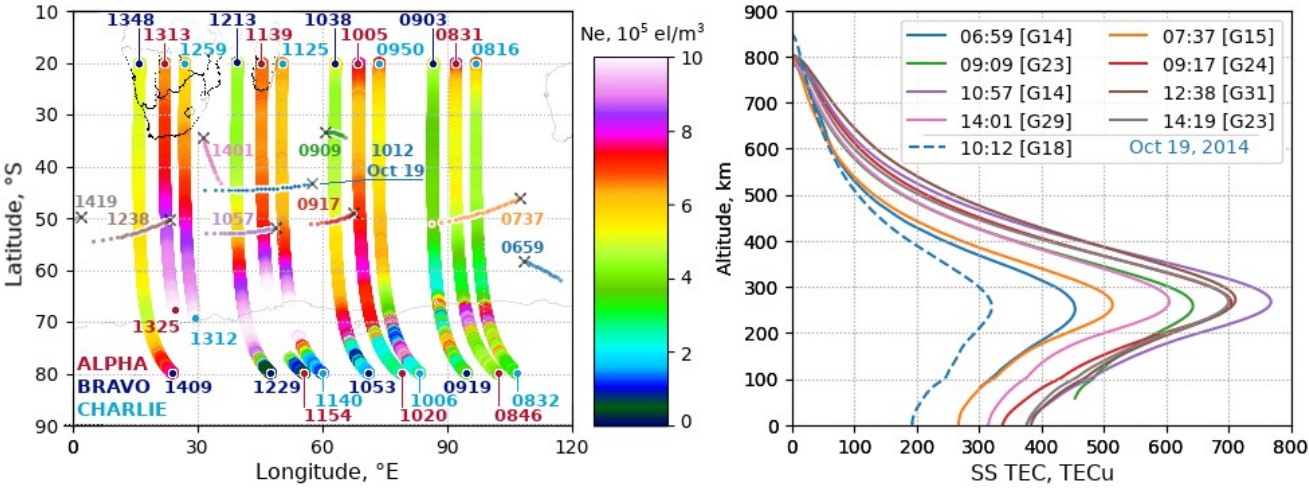

**Figure 2: Electron concentration during 8-14 UT of April 5, 2014 measured by SWARM (left) and total electron content from RO measurements by COSMIC satellites (right, solid). Compare with the profile of October 19, 2014 when no LTE was detected (right, dashed). Digits marks time of observation in a format HHMM (hours and minutes)**

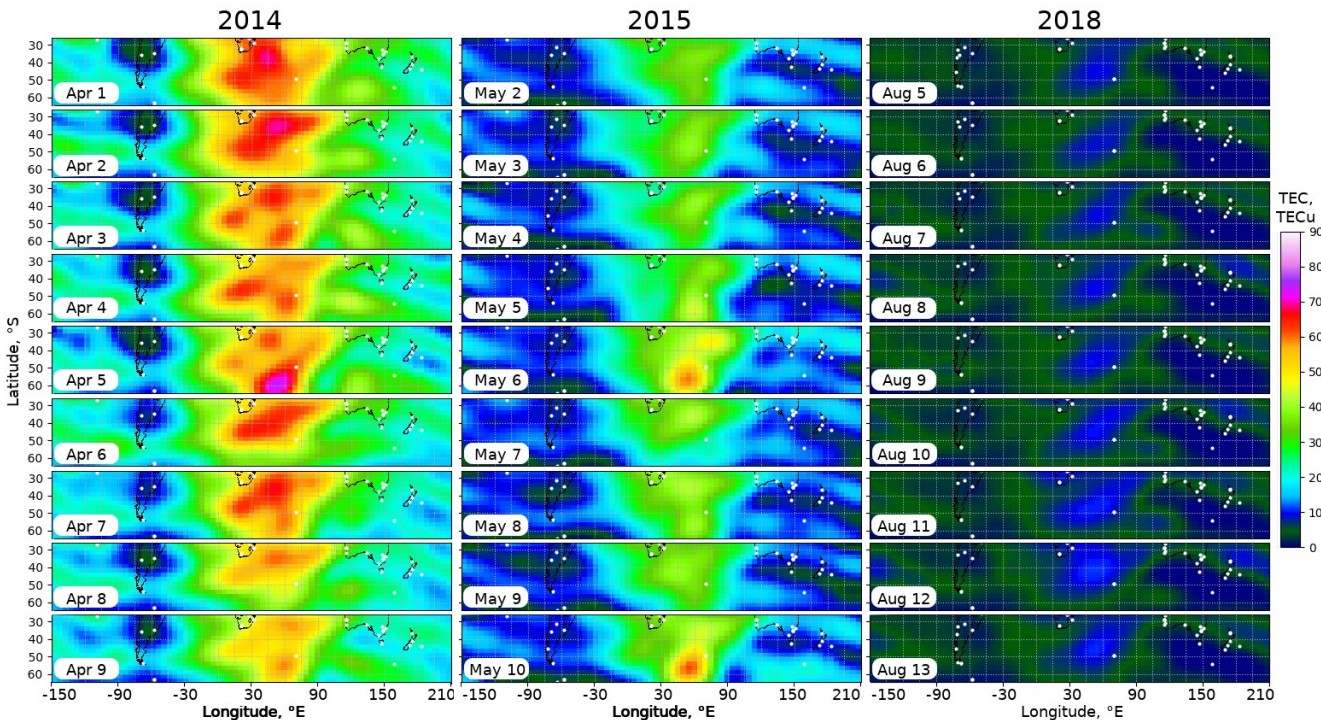

**Figure 3: Series of LTEs observed in Southern Hemisphere at 10 UT during years of different solar activity**

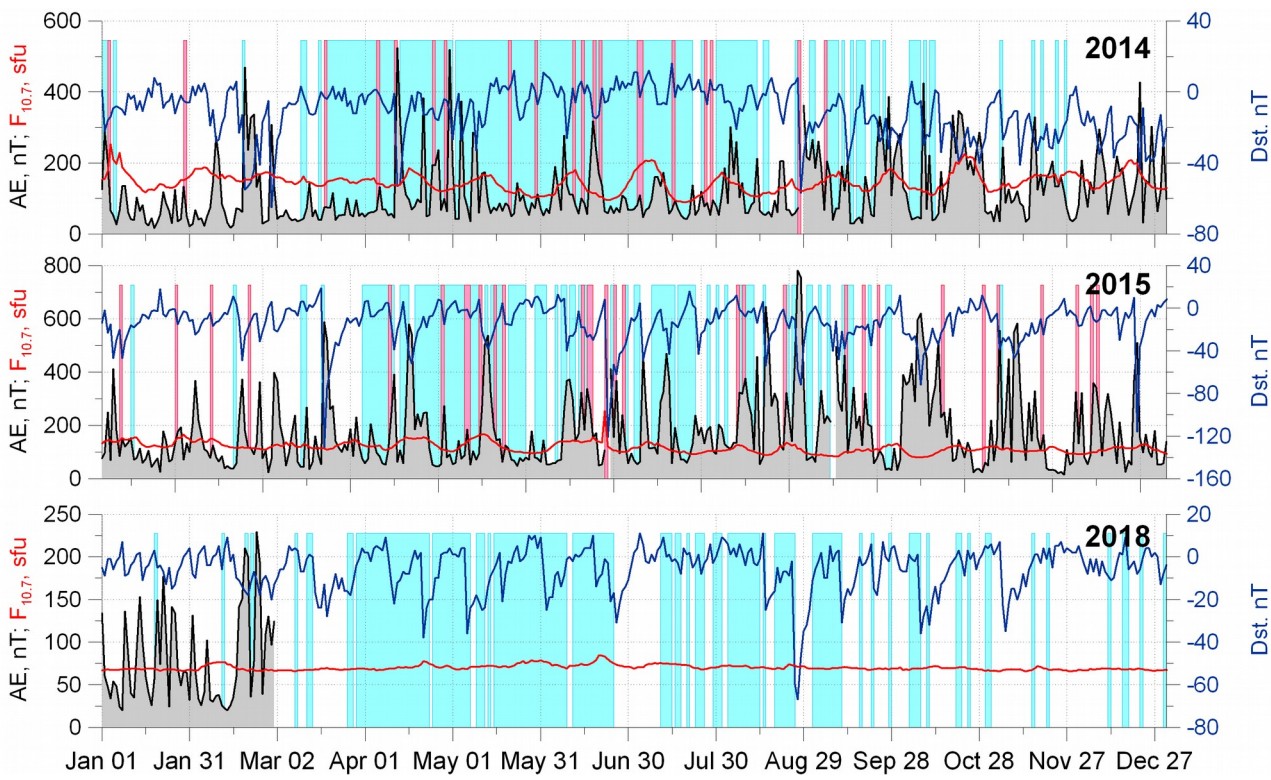

**Figure 4: Days with the LTE observed in SH (blue bars) during 2014 (top), 2015 (middle) and 2018 (bottom). Plots show annual variations of daily average values of $F_{10.7}$ (red), AE (black, filled with gray) and Dst (navy blue) indices. Cases of bright SLTE observations are highlighted by red bars. In 2018 AE index is available only for Jan and Feb.**


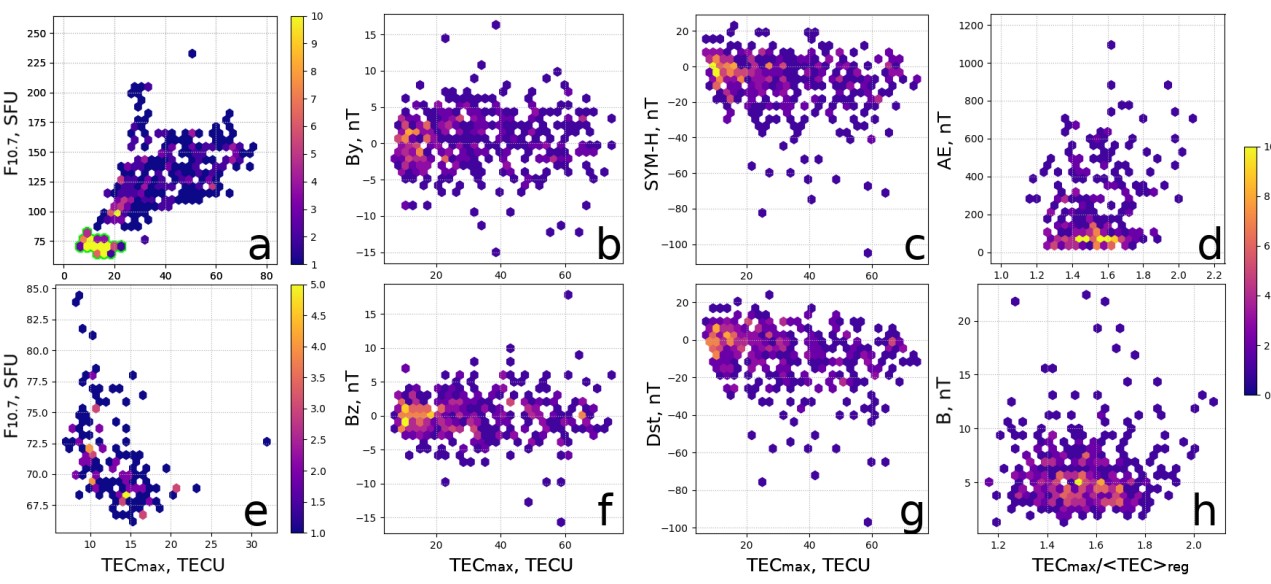

**Figure 5: Distributions of maximal TEC values in a region 30°W-60°E, 30-60°S versus 10.7 nm solar radiation (a, e), components of IMF By (b) and Bz (f), geomagnetic indices SYM-H (c) and Dst (g);  and distributions of maximal to regional mean TEC ratio versus Auroral electrojet index AE (d) and IMF intensity B (h). All the TEC values are taken for 10 UT during years 2014, 2015 and 2018. Distribution in (e) is made with data only for 2018; this data is highlighted with green in panel (a).**

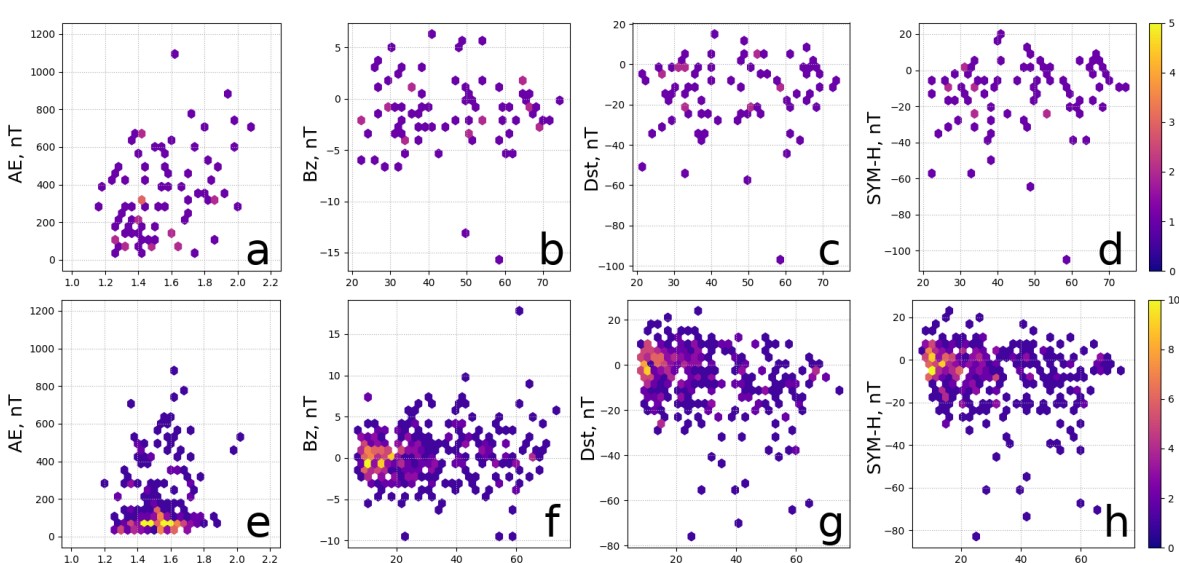

**Figure 6. Distributions of TECratio and TECmax versus main geomagnetic indices for days with high intensity SLTE (top) and for all the others LTE detected over 2014, 2015, 2018.**

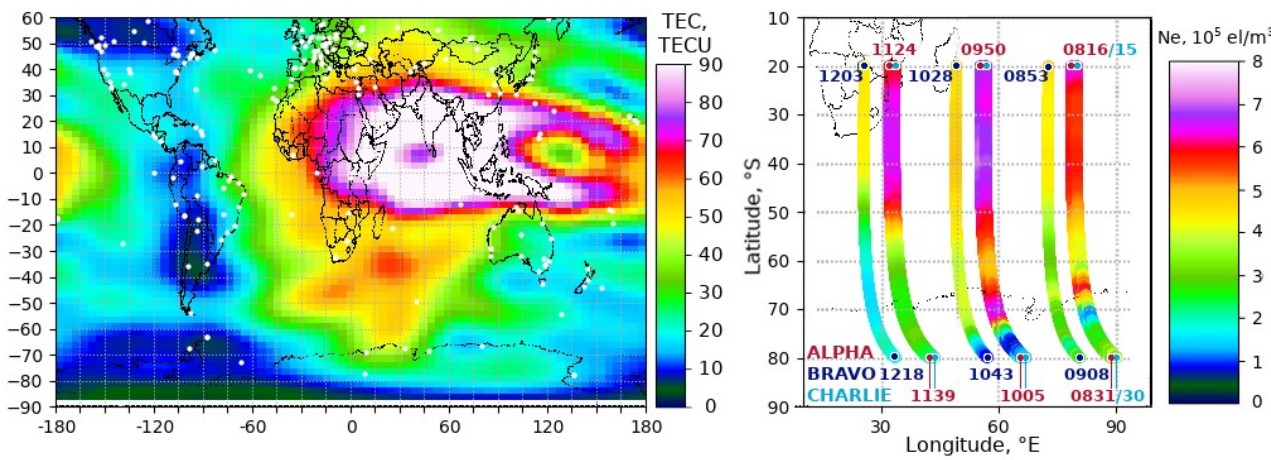

**Figure 7. LTE of Apr 18, 2014 observed in GIM (left) and in-situ measurements of Ne by SWARM satellites (right)**

