# Peer review of "Localized TEC enhancements in the Southern Hemisphere"

_Annales Geophysicae, 2019_

## Referee Comment (RC1) · Anonymous Referee #1 · 22 Oct 2019

Review of "Localized TEC enhancements in the Southern Hemisphere" by Edemskiy

The goal of this paper is to reveal the main morphology of localized TEC (total electron content) enhancements (LTEs), particularly of LTE series, detected in the Southern Hemisphere using global ionospheric maps for different solar activity years (2014, 2015, 2018). I believe the paper is not close to being acceptable for publication in Annales Geophysicae in its present form. In my opinion several main points should be considered and clarified before publication.

1. Authors mentioned (lines 20-22) that: "The Southern Hemisphere contains at least two large anomalous regions: South Atlantic Magnetic Anomaly and Weddell Sea Anomaly. The latter consists in the modulation of TEC's diurnal oscillations by the solar‐modulated seasonal oscillations, which produces a diurnal anomaly in the

vicinity of the Weddell Sea during Southern Hemisphere summer (October to March) (Lean et al., 2016)." I recommend author to take the traditional (more clear) definition of WSA phenomenon.

2. (Lines 34-35): "During analysis of ionosphere response to a geomagnetic storm of 15 August 2015, a curious structure was detected in global ionospheric maps (GIMs), which we call localized TEC enhancement or LTE (Edemskiy et al., 2018)." The term localized TEC enhancement was mentioned many years before by Foster and Rideout (2007) and Foster and Coster (2007). Note that Foster et al. studies present localized TEC enhancement in Northern hemisphere many times. I recommend author to read John Foster's et al. articles in order to understand the morphology and physical explanation of localized TEC enhancement in NH. I believe that these papers should give you new information.

J. C. Foster, W. Rideout. Storm enhanced density: magnetic conjugacy effects. Annales Geophysicae, 2007, 25 (8), pp.1791-1799. Foster, J. C. and Coster, A. J.: Localized stormtime enhancement of TEC at Low Latitudes in the american sector, J. Atmos. Solar-Terr. Phys., 69, 1241–1252, doi:10.1016j.jastp.2006.09.012, 2007.

3. (Lines 45-55). Unfortunately there are many remarks about definition of LTE's. According to first sentence here "The localized TEC enhancement is a positive disturbance of ionosphere." But according to two detection criteria the LTE is a spatial-temporal structure in the UT map of TEC and is not a disturbance.

4. (Lines 47-48): "1. Spatial limitation and clear borders. An enhancement should not be wider than 40° and 120° in latitude and longitude, respectively. Gradients at an LTE edges should be high enough to make LTE borders possible to distinguish." According to such limitation almost all of winter UT map of TEC should reveal LTE due to 1) short duration (therefore limitation in longitude smaller then 90°) and significant gradients of TEC diurnal variation during sunlight hours; 2) clear border at sub-auroral latitudes due to pronounce main ionospheric trough structure (for daytime also). So

according to these criteria, I don't understand how LTE can be distinguished from the usual TEC maps in winter and equinox seasons. Figure 3 demonstrate many cases of consistence between LTE and typical TEC diurnal variation (that is presented in view of longitude-latitude map for UT epoch). Another problem is statement: "Gradients at an LTE edges should be high enough". Please provide mathematical formulation for "should be high enough". Or this criteria was checked manually for each maps?

5. (Lines 52-54): "We search LTEs only in the Southern Hemisphere, because Edemskiy et al. (2018) detected LTEs only at SH. A disturbance should follow the Sun having the maximal intensity no latter than 1-2 hours after local noon (in a period 12-14 LT as observed by Edemskiy et al., 2017). I disagree with argument for LTE limitation only in the Southern Hemisphere. Edemskiy et al. (2018) study concern to geomagnetic storm response on particular event on Aug 2017. There are many examples of daytime storm-time localized TEC enhancement (Foster and Rideout, 2007; Zhao et al., 2012) in NH. Why author's algorithm exclude all these situations? I did not found Edemskiy et al., 2017 in the reference list.

6. (Lines 89-92): "An example of a clearly observed LTE was detected at April 5, 2014 (Fig. 1). The disturbance reached the highest intensity in a period 10-12 UT when TEC values in a center of the disturbance exceeded 78 TECU. This value is comparable to equatorial TEC values. The highest values were detected in a latitudinal region 45-70°S. At the same time, TEC values of the entire region (30-70°S, 0-90°E) were enhanced." The reason for SLTE is associated to geomagnetic disturbances during 5 April. Please see AE index on Fig. 1. It is evident that geomagnetic disturbances in AE started at 06 UT on 5 April, 2014 (the same as SLTE). The maximal geomagnetic disturbance occur at 10-12 UT. At the same time the highest intensity of SLTE occur when TEC values in a center of the disturbance exceeded 78 TECU. So SLTE in reality can be SED structure or something else that associated with geomagnetic disturbances.

7. (Lines 94-95): "As it will be shown later, such a strong SLTE is not typical and in some cases it is not detected at all." Why author to select this case if this case is not

typical?

8. (Lines 106-107): "In-situ measurement of electron concentration Ne from SWARM satellites allow us to check validity of TEC distribution presented by GIM." In my own opinion Fig. 2 provide clear evidence of SLTE, but not for MLTE. So according to my points 6-8 Figs. 1 and 2 does not give to reader typical examples of LTE. I recommend to add a typical example of MLTE that does not associated with geomagnetic disturbances.

9. Figure 5. IMF intensity is not a good choice of parameter that determine geomagnetic activity because direction of IMF Bz can be more important for ionospheric disturbances. In my opinion AE or AP index can be more effective in this investigation.

10. About discussion part. It is very chaotic. I still did not understand which of the mechanisms, according to the author, is the main one for the formation of LTE.

There are a lot of additional questions according to LTE, but I stopped here in order to obtain some clarification about LTE.

Please also note the supplement to this comment:
https://www.ann-geophys-discuss.net/angeo-2019-124/angeo-2019-124-RC1-supplement.pdf

———————————————————

[Figure]

[Figure]

**Fig. 1.**

---

## Referee Comment (RC2) · Anonymous Referee #2 · 7 Nov 2019

MS No.: angeo-2019-124

Title: Localized TEC enhancements in the Southern Hemisphere

Author(s): Ilya K. Edemskiy

The paper presents the results of registering localized TEC enhancements (LTEs) in the Southern Hemisphere. In my opinion, the author has detected interesting phenomenon that has received no attention before. However, some aspects should be refined. The paper is suitable by topic for Annales Geophysicae and may be accepted for publication after some revision.

General Comments

1. It is not clear from the text what the author regards as a possible mechanism of LTE generation. A possible mechanism of LTE generation should be one of the main conclusions in section 4. The author is obviously not yet able to indicate the exact mechanism. However, he should single out and discuss possible mechanisms.

2. As I believe (see Comment for Line 163), the author uses the intensity of the interplanetary magnetic field (IMF) to analyze LTE dependence on geomagnetic activity level. But for this analysis, it is better to use the magnetic activity indices Dst (or SYM-H) and AE, which make it easy to select disturbance periods in Earth's magnetic field variations. In addition to Figures 4 and 5, it would be useful to add a figure to show the time variations of Dst, AE, F10.7 and "temporal position" of each LTE during all the years (2014, 2015, 2018).

3. Throughout the text, please check the season names in the Southern Hemisphere: in some places March-July are called "autumn-winter" (Lines 143-144, 187-188), and in other places they are referred to as "spring-summer" (Lines 148-149, 227-228).

Comments

Lines 5-6. In Abstract, it is not clear what the author means by "LTE series". Please keep in mind that a lot of people read Abstract only.

Lines 7-8. "It is shown that LTE intensity varies in dependence on solar flux and does not directly depend on interplanetary magnetic field orientation." LTE dependence on interplanetary magnetic field orientation is not discussed in the paper. See also Comments for Line 231.

Lines 19-25. "The Southern Hemisphere contains at least two large anomalous regions: South Atlantic Magnetic Anomaly and Weddell Sea Anomaly." Since the author mentions two large anomalous regions in the Southern Hemisphere (South Atlantic Magnetic Anomaly and Weddell Sea Anomaly), he should characterize both of them, not one (Weddell Sea Anomaly).

Moreover, I would recommend to pay particular attention to the South Atlantic Magnetic Anomaly (SAMA). The SAMA region is very close to the area where LTEs are detected (Fig. 1A and Fig. 3). Perhaps SAMA (itself or together with some other factors, such as a neutral wind, for example) promotes the LTE formation.

On the other hand, LTE looks like a continuation of the region occupied by the Equatorial Ionization Anomaly (EIA) in Fig. 1A (unfortunately, the boundaries of Fig. 3 cut off the EIA, and nothing can be said here). Maybe sometimes one get conditions that allow a plume from EIA "fountain" to reach higher latitudes.

Lines 32-33. "However, none of these models predict the occurrence of the LTE phenomenon." Neither the abbreviation "LTE" nor the term "LTE" have been used before. Please, explain what "LTE" is before using it. In a scientific article, one should avoid term/abbreviation explanations after their first use. This makes understanding difficult.

Line 48. "Gradients at an LTE edges should be high enough to make LTE borders possible to distinguish." Please, specify the numerical value of the gradient threshold you use.

Line 54. "Edemskiy et al., 2017" Probably, the author meant "Edemskiy et al., 2018"

Lines 125-126. "Blue dashed line (Fig. 2, right) presents a profile measured at 10:12 UT at October 19, 2014 when there was no LTE observed in GIM." Please, explain why October 19, 2014 was chosen as a day without LTE. Though April days with LTE are analyzed. Why did not you use a day without LTE closer to April?

Lines 133-134. "The intensity and the shape of the presented LTEs vary but at the same time of day all of them occupy the same region." should be replaced with "The intensity and shape of the presented LTEs vary from day to day, but at the same time of day all of the LTEs occupy the same region".

Lines 137-138. "In a similar way LTE series were observed during other investigated years of relatively high (2015) and low (2018) solar activity." It is necessary to clarify

what the level of solar activity was in 2014 and what index was used for the solar activity characteristic. The author should also indicate numerical values of the solar activity level for each year. Please, explain what "LTE series" is.

Line 163. "distribution of this ratio vs IMF intensity." Please, explain: - what "IMF" is; - what "IMF intensity" is: whether it is B intensity or Bz intensity;

Line 178. Article [Cherniak et al., 2012] is not included in References.

Line 225. "5 Discussion" Probably, the author meant "5 Summary".

Line 231. "No clear dependence between orientation of IMF and LTEs' parameters was observed." LTE dependence on interplanetary magnetic field orientation is not discussed in the text. Therefore, this conclusion is not substantiated.

Lines 240-305, References. Articles [Afonin et al., 1995], [Chen et al., 2011], [He et al., 2011], [Krankowski, et al., 2009], [Matyjasiak, et al., 2005], [Sun, et al., 2017] are not mentioned in the text.

---

## Author Comment (AC1) · 21 Dec 2019

1. Authors mentioned (lines 20-22) that: "The Southern Hemisphere contains at least two large anomalous regions: South Atlantic Magnetic Anomaly and Weddell Sea Anomaly. The latter consists in the modulation of TEC's diurnal oscillations by the solar-modulated seasonal oscillations, which produces a diurnal anomaly in the vicinity of the Weddell Sea during Southern Hemisphere summer (October to March) (Lean et al., 2016)." I recommend author to take the traditional (more clear) definition of WSA phenomenon.

—

This part of introduction was changed: "It is known that Southern Hemisphere contains

some anomalous regions. South Atlantic Magnetic Anomaly (SAMA) is formed by a configuration of geomagnetic field which has a global minimum of intensity over South Atlantic and South America and makes it easier for energetic particles of inner radiation belt to precipitate, thus increasing ionospheric conductivity over the region (Abdu et al., 2005). South of the SAMA, in the south-eastern Pacific and South Atlantic Antarctic regions, combination of the geomagnetic field features and thermospheric winds produces an inverted diurnal plasma density pattern at equinoxes and in SH summer (October-March): the nighttime maximum is larger than the daytime minimum, and the phenomenon is known as the Weddell Sea Anomaly (WSA) (Horvath, 2006). Jakowski et al. (2015) showed that during periods of low solar activity in Asian longitudinal sector of SH it is possible to observe so called nighttime winter anomaly (NWA), when values of electron concentration are higher in winter that in summer. At the same time Yasyukevich et al. (2018) showed that winter anomaly manifests itself much less intensively in SH that in NH. It is possible to conclude that ionosphere of each hemisphere has some specific features."

2. (Lines 34-35): "During analysis of ionosphere response to a geomagnetic storm of 15 August 2015, a curious structure was detected in global ionospheric maps (GIMs), which we call localized TEC enhancement or LTE (Edemskiy et al., 2018)." The term localized TEC enhancement was mentioned many years before by Foster and Rideout (2007) and Foster and Coster (2007). Note that Foster et al. studies present localized TEC enhancement in Northern hemisphere many times. I recommend author to read John Foster's et al. articles in order to understand the morphology and physical explanation of localized TEC enhancement in NH. I believe that these papers should give you new information.

—

Thank you for your recommendation. The text was changed and reference to Foster et al. papers was added. At the same time it should be noted that their papers mostly describe ionosphere during geomagnetic storms, investigating manifestation of storm

enhanced density (SED), whereas the considered LTEs are observed almost independently on geomagnetic conditions, even during quiet periods (Kp=1). The introduction was changed and the following was added: "The most typical irregularities in distribution of electron concentration are produced during geomagnetic storms. Foster and Coster (2007) investigating storm enhanced densities (SED). They showed that during severe and extreme storms it is possible to detect SEDs which in maps of total electron content (TEC) could be observed as localized TEC enhancements (LTE). The authors showed that during a storm recovery phase LTEs could be detected in the night side ionosphere at the middle latitudes of both hemispheres, in magneto-conjugated regions. The authors note that the observed enhancements are approximately corotating in place over the positions in which they were formed earlier in the event. However, the LTE phenomenon studied by Foster and Coster (2007) is different from the LTE phenomenon studied by us. During analysis of ionospheric response to a geomagnetic storm of 15 August 2015 Edemskiy et al. (2018) detected a curious LTE in global ionospheric maps (GIMs). Unlike the LTEs observed by Foster and Coster (2007) this enhancement was detected and observed in sunlit area of the Southern Hemisphere and lasted for several hours. It was not corotating but changing position following the Sun and propagating along the geomagnetic parallels. Using quite a simple detection algorithm Edemskiy et al. (2018) found about 30 similar events in the Southern Hemisphere during 2010-2016 and some of the detected LTEs were observed during relatively quiet periods. The authors showed direct dependence of number of the detected LTEs on solar activity level and suggested that their generation is connected with the orientation of interplanetary magnetic field (IMF), namely with Bz."

3. (Lines 45-55). Unfortunately there are many remarks about definition of LTE's. According to first sentence here "The localized TEC enhancement is a positive disturbance of ionosphere." But according to two detection criteria the LTE is a spatial-temporal structure in the UT map of TEC and is not a disturbance.

—

Yes, thank you, you are absolutely right. Now it is noted in the text that LTE should not be considered as a disturbance. The criteria were changed as well: "In this paper a TEC enhancement is considered as LTE if it is: - located in middle latitudes of sunlit region. Mainly we investigate LTEs, which are clearly observed in Indian and Southern part of Atlantic Oceans and do not take into account enhancements in Northern Hemisphere. At the same time, LTEs in SH are not accompanied by any LTE in NH and fuch a focusing on SH LTEs is quite reasonable. - spatially limited by relatively lower TEC values. Normalized difference between squared maximal value in LTE and minimal one at its border ($\Delta=1$ - (Iedge/Imax)ˆ2) should be no less than 20%. Generally that means that there should be a clear trough between an enhancement and the equatorial ionization anomaly (EIA). - confined and have a border of lower TEC values ($\Delta \geq 20\%$) no farther than in $40°$ in longitude from the location of maximal TEC value. Mainly that means that we do not consider longitudinally stretched enhancements assuming different mechanism of their generation."

4. (Lines 47-48): "1. Spatial limitation and clear borders. An enhancement should not be wider than 40 âŮę and 120 âŮę in latitude and longitude, respectively. Gradients at an LTE edges should be high enough to make LTE borders possible to distinguish." According to such limitation almost all of winter UT map of TEC should reveal LTE due to 1) short duration (therefore limitation in longitude smaller then 90 âŮę ) and significant gradients of TEC diurnal variation during sunlight hours; 2) clear border at sub-auroral latitudes due to pronounce main ionospheric trough structure (for daytime also). So according to these criteria, I don't understand how LTE can be distinguished from the usual TEC maps in winter and equinox seasons. Figure 3 demonstrate many cases of consistence between LTE and typical TEC diurnal variation (that is presented in view of longitude-latitude map for UT epoch). Another problem is statement: "Gradients at an LTE edges should be high enough". Please provide mathematical formulation for "should be high enough". Or this criteria was checked manually for each maps?

—

Thank you for this remark. The criteria definition was changed to describe an LTE more precisely. Key point which was missed previously is a presence of trough between EIA and the observed enhancement. Demands on the minimal depth of this trough is given mathematically. All the enhancements in fig. 3 fulfill the criteria and are considered as LTE. Several examples of maps showing other types of enhancement or absence of enhancement are available via the link below. Such situations were considered as non-LTE cases and were not considered during the investigation. https://drive.google.com/drive/folders/1u6GTyRe9bIFb-Kb25gMKkGM5_LIvbQAv?usp=sharing

5. (Lines 52-54): "We search LTEs only in the Southern Hemisphere, because Edemskiy et al. (2018) detected LTEs only at SH. A disturbance should follow the Sun having the maximal intensity no latter than 1-2 hours after local noon (in a period 12-14 LT as observed by Edemskiy et al., 2017). I disagree with argument for LTE limitation only in the Southern Hemisphere. Edemskiy et al. (2018) study concern to geomagnetic storm response on particular event on Aug 2017. There are many examples of daytime storm-time localized TEC enhancement (Foster and Rideout, 2007; Zhao et al., 2012) in NH. Why author's algorithm exclude all these situations? I did not found Edemskiy et al., 2017 in the reference list.

—

Misprint "Edemskiy et al., 2017" is corrected: "Edemskiy et al., 2018" The statement was badly formulated. The point was that this particular paper is dedicated only to LTEs detected in Southern Hemisphere, particularly in South Indian and South Atlantic regions. The same LTE was described by Edemskiy et al. (2018) and the idea of the current paper is to find other similar enhancements using GIMs. During the investigation it was found that these structures can be detected not only during magnetic storms but during quiet days as well. The author does not claim absence of LTEs in Northern Hemisphere or in other ranges of longitude. However during SH LTE we do not see any corresponding effect in NH. At the same time the aim was not to describe only stormtime LTE, but any such formation which are observed only in Southern Hemisphere. This is reflected in new formulation of the criteria.

6. (Lines 89-92): "An example of a clearly observed LTE was detected at April 5, 2014 (Fig. 1). The disturbance reached the highest intensity in a period 10-12 UT when TEC values in a center of the disturbance exceeded 78 TECU. This value is comparable to equatorial TEC values. The highest values were detected in a latitudinal region 45-70 âŮę S. At the same time, TEC values of the entire region (30-70 âŮę S, 0-90 âŮę E) were enhanced." The reason for SLTE is associated to geomagnetic disturbances during 5 April. Please see AE index on Fig. 1. It is evident that geomagnetic disturbances in AE started at 06 UT on 5 April, 2014 (the same as SLTE). The maximal geomagnetic disturbance occur at 10-12 UT. At the same time the highest intensity of SLTE occur when TEC values in a center of the disturbance exceeded 78 TECU. So SLTE in reality can be SED structure or something else that associated with geomagnetic disturbances.

—

Thank you very much for this remark. It moved the investigation forward. SED structures are mentioned both in introduction and discussion sections and briefly described in the latter. The possible connection between SED and LTE is discussed and some statistical analyses are added as well. At the same time it is shown that clear SLTE are observed during relatively quiet conditions with positive Dst and small AE values; and otherwise: not all storms were accompanied by SLTEs.

7. (Lines 94-95): "As it will be shown later, such a strong SLTE is not typical and in some cases it is not detected at all." Why author to select this case if this case is not typical?

—

This day was chosen since despite the more intense SLTE both the enhancements are

observed quite clearly. This structure fulfills the criteria of LTE and at least partially can be clearly seen in SWARM data. Another map with a LTE confirmed by SWARM is added to the discussion.

8. (Lines 106-107): "In-situ measurement of electron concentration Ne from SWARM satellites allow us to check validity of TEC distribution presented by GIM." In my own opinion Fig. 2 provide clear evidence of SLTE, but not for MLTE. So according to my points 6-8 Figs. 1 and 2 does not give to reader typical examples of LTE. I recommend to add a typical example of MLTE that does not associated with geomagnetic disturbances.

—

Thank you for this remark. Another figure containing MLTE and corresponding SWARM measurements are added to the paper. Some text clarifying this problem was added to the discussion section.

9. Figure 5. IMF intensity is not a good choice of parameter that determine geomagnetic activity because direction of IMF Bz can be more important for ionospheric disturbances. In my opinion AE or AP index can be more effective in this investigation.

—

Analysis of LTE occurrence rate dependence on geomagnetic indices or IMF components did not reveal any trends that is why these distributions were not shown in figures. The dependence on Bz was the initial hypothesis which was not supported by observations. The fig. 5 is changed now and contains all the basic geomagnetic indices.

10. About discussion part. It is very chaotic. I still did not understand which of the mechanisms, according to the author, is the main one for the formation of LTE. There are a lot of additional questions according to LTE, but I stopped here in order to obtain some clarification about LTE.
[Figure]

—

The discussion part was almost fully rewritten and now contains more details and suggestions about LTEs.

---

## Author Comment (AC2) · 21 Dec 2019

I thank both reviewers for helpful comments.

1. It is not clear from the text what the author regards as a possible mechanism of LTE generation. A possible mechanism of LTE generation should be one of the main conclusions in section 4. The author is obviously not yet able to indicate the exact mechanism. However, he should single out and discuss possible mechanisms.

Discussion section was almost fully rewritten and now contain some suggestions about the mechanism

2. As I believe (see Comment for Line 163), the author uses the intensity of the interplanetary magnetic field (IMF) to analyze LTE dependence on geomagnetic activity

level. But for this analysis, it is better to use the magnetic activity indices Dst (or SYM-H) and AE, which make it easy to select disturbance periods in Earth's magnetic field variations. In addition to Figures 4 and 5, it would be useful to add a figure to show the time variations of Dst, AE, F10.7 and "temporal position" of each LTE during all the years (2014, 2015, 2018).

Thank you for the recommendation. The figure 4 is replaced by plots showing temporal position of each LTE and the corresponding values of F10.7 and SYM-H indices. Speaking of IMF intensity usage for the comparison, figure 5 is updated and now presents dependencies on AE, Dst and SYM-H as well.

3. Throughout the text, please check the season names in the Southern Hemisphere: in some places March-July are called "autumn-winter" (Lines 143-144, 187-188), and in other places they are referred to as "spring-summer" (Lines 148-149, 227-228).

Thank you. This confusion in the names was corrected.

Thank you for all the comments. They really helped to improve the paper

Comments

Lines 5-6. In Abstract, it is not clear what the author means by "LTE series". Please keep in mind that a lot of people read Abstract only.

I changed the Abstract in accordance with the comment

Lines 7-8. "It is shown that LTE intensity varies in dependence on solar flux and does not directly depend on interplanetary magnetic field orientation." LTE dependence on interplanetary magnetic field orientation is not discussed in the paper.

Line 231. "No clear dependence between orientation of IMF and LTEs' parameters was observed." LTE dependence on interplanetary magnetic field orientation is not discussed in the text. Therefore, this conclusion is not substantiated.

The distributions similar to ones in fig. 5 were calculated but were not presented since

they do not show any trend or dependence of LTE occurrence. The figure was changed to show distributions versus all the main parameters and the corresponding text was added.

Lines 19-25. "The Southern Hemisphere contains at least two large anomalous regions: South Atlantic Magnetic Anomaly and Weddell Sea Anomaly." Since the author mentions two large anomalous regions in the Southern Hemisphere (South Atlantic Magnetic Anomaly and Weddell Sea Anomaly), he should characterize both of them, not one (Weddell Sea Anomaly).

Moreover, I would recommend to pay particular attention to the South Atlantic Magnetic Anomaly (SAMA). The SAMA region is very close to the area where LTEs are detected (Fig. 1A and Fig. 3). Perhaps SAMA (itself or together with some other factors, such as a neutral wind, for example) promotes the LTE formation.

On the other hand, LTE looks like a continuation of the region occupied by the Equatorial Ionization Anomaly (EIA) in Fig. 1A (unfortunately, the boundaries of Fig. 3 cut off the EIA, and nothing can be said here). Maybe sometimes one get conditions that allow a plume from EIA "fountain" to reach higher latitudes.

The description of the SH anomalies is corrected and now contains more details about each anomaly. Obviously the presented LTEs are connected with configuration of geomagnetic field as well as SAMA anomaly, but it should be noted that the last one is mostly located in Atlantic ocean, when LTE is typically generated and develops in Indian ocean and in geomagnetic latitudes which are usually higher than those of SAMA.

Speaking of continuation of EIA, there are several things to be noted:

- we observe SH LTEs asymmetrically: independent on the season there are no similar structures in NH even during equinox periods, when amount of solar radiation is quite the same in both hemispheres. And it is not clear why we do not have the same continuation in NH.

- being observed in near-noon area LTE should be formed by solar ionization which impact is maximal in sub-equatorial region.

- It is possible to observe enhancements which are continuation of EIA (e.g. a figure below) and we do not consider them as LTE since they are not localized (do not have clear border)

The figure below shows enhanced values in SH, which do not fulfill the criteria and looks like a continuation of EIA.

Lines 32-33. "However, none of these models predict the occurrence of the LTE phenomenon." Neither the abbreviation "LTE" nor the term "LTE" have been used before. Please, explain what "LTE" is before using it. In a scientific article, one should avoid term/abbreviation explanations after their first use. This makes understanding difficult.

The abbreviation was introduced in abstract, but I agree that it should be introduced in the text as well. A short explanation is added.

Line 48. "Gradients at an LTE edges should be high enough to make LTE borders possible to distinguish." Please, specify the numerical value of the gradient threshold you use.

The formulation of the criteria was changed:

"In this paper a TEC enhancement is considered as LTE if it is:

- located in middle latitudes of sunlit region. Mainly we investigate LTEs, which are clearly observed in Indian and Southern part of Atlantic Oceans and do not take into account enhancements in Northern Hemisphere. At the same time, LTEs in SH are not accompanied by any LTE in NH and fuch a focusing on SH LTEs is quite reasonable.

- spatially limited by relatively lower TEC values. Normalized difference between squared maximal value in LTE and minimal one at its border ($\Delta=1 - (I\_edge/I\_max)^2$) should be no less than 20%. Generally that means that there should be a clear trough

between an enhancement and the equatorial ionization anomaly (EIA).

- confined and have a border of lower TEC values (△≥20%) no farther than in 40° in longitude from the location of maximal TEC value. Mainly that means that we do not consider longitudinally stretched enhancements assuming different mechanism of their generation."

Line 54. "Edemskiy et al., 2017" Probably, the author meant "Edemskiy et al., 2018"

Yes, that was a misprint

Lines 125-126. "Blue dashed line (Fig. 2, right) presents a profile measured at 10:12 UT at October 19, 2014 when there was no LTE observed in GIM." Please, explain why October 19, 2014 was chosen as a day without LTE. Though April days with LTE are analyzed. Why did not you use a day without LTE closer to April?

The problem consists of two parts: COSMIC should be at a proper position near the local noon to make it possible to observe the given area in Southern Hemisphere; and we did not observe enhanced TEC at this moment. As it could be seen from fig. 4, there was the only day without LTE (Apr 12) and during this day TEC values were enhanced but not fulfilled the LTE criteria. A corresponding short explanation is added to the text:

"Due to the phenomenon of LTE series which will be described later, TEC values over the given region are enhanced almost during the whole month. To demonstrate iono-sphere profile without any enhancement in GIM we have chosen October 19, 2014. The profile measured by COSMIC at 10:12UT is shown by dashed blue line in fig. 2."

Lines 133-134. "The intensity and the shape of the presented LTEs vary but at the same time of day all of them occupy the same region." should be replaced with "The intensity and shape of the presented LTEs vary from day to day, but at the same time of day all of the LTEs occupy the same region".

Your text is better, thank you. Your formulation is used in the text.

Lines 137-138. "In a similar way LTE series were observed during other investigated years of relatively high (2015) and low (2018) solar activity." It is necessary to clarify what the level of solar activity was in 2014 and what index was used for the solar activity characteristic. The author should also indicate numerical values of the solar activity level for each year. Please, explain what "LTE series" is.

Such a definition of LTE series is given in text:

"We define such a continuous sequence of LTEs observed day by day as a series of LTE. At least two consequently observed LTEs are considered as a series."

The following text was added to the section Data and methods.

"The estimation of solar activity level is based on F10.7 index from OMNI database. During the investigated years monthly averaged F10.7 values were varying in ranges 130-160 (2014), 95-135 (2015), 65-75 (2018) sfu corresponding to high, relatively high and low level of solar activity. "

Line 163. "distribution of this ratio vs IMF intensity." Please, explain: - what "IMF" is; -what "IMF intensity" is: whether it is B intensity or Bz intensity;

IMF intensity was standing for B intensity. All the mentioned terms "IMF intensity" are replaced by "IMF intensity B".

Line 178. Article [Cherniak et al., 2012] is not included in References.

The missed reference was added

Line 225. "5 Discussion" Probably, the author meant "5 Summary".

The misprint is corrected to "5 Conclusions"

Lines 240-305, References. Articles [Afonin et al., 1995], [Chen et al., 2011], [He et al., 2011], [Krankowski, et al., 2009], [Matyjasiak, et al., 2005], [Sun, et al., 2017] are not mentioned in the text.

Unused references were removed
[Figure]

**codg [Z] 2014-10-28 10:00:00**

**Fig. 1.** Example of EIA continuation

[Figure]

---

## Author Comment (AC4) · 21 Dec 2019

The paper is available via the link:

https://drive.google.com/drive/folders/1BbxbtGyK67uUHe4RbYhDrctGp9ojxfsR?usp=sharing

---

## Author Comment (AC5) · 21 Dec 2019

The edited paper is available via the link:

https://drive.google.com/drive/folders/1BbxbtGyK67uUHe4RbYhDrctGp9ojxfsR?usp=sharing

---

## Author Response (AR2)

I am very grateful to the reviewer for the kind report

1. Lines 111-112: Generally that means that there should be a clear trough between enhancement and the equatorial ionization anomaly (EIA).
Do you mean trough between LTE and EIA southern crest? Please re-phrase.

The formulation was changed to:
Generally that means that between southern crest of equatorial ionization anomaly (EIA) and a region of enhanced TEC there should be observed a clear trough.